# Learning towards Minimum Hyperspherical Energy

**Weiyang Liu[1,*], Rongmei Lin[2,*], Zhen Liu[1,*], Lixin Liu[3], Zhiding Yu[4], Bo Dai[1,5], Le Song[1,6]**
[1]Georgia Institute of Technology   [2]Emory University
[3]South China University of Technology   [4]NVIDIA   [5]Google Brain   [6]Ant Financial

## Abstract

Neural networks are a powerful class of nonlinear functions that can be trained end-to-end on various applications. While the over-parametrization nature in many neural networks renders the ability to fit complex functions and the strong representation power to handle challenging tasks, it also leads to highly correlated neurons that can hurt the generalization ability and incur unnecessary computation cost. As a result, how to regularize the network to avoid undesired representation redundancy becomes an important issue. To this end, we draw inspiration from a well-known problem in physics – Thomson problem, where one seeks to find a state that distributes $N$ electrons on a unit sphere as evenly as possible with minimum potential energy. In light of this intuition, we reduce the redundancy regularization problem to generic energy minimization, and propose a minimum hyperspherical energy (MHE) objective as generic regularization for neural networks. We also propose a few novel variants of MHE, and provide some insights from a theoretical point of view. Finally, we apply neural networks with MHE regularization to several challenging tasks. Extensive experiments demonstrate the effectiveness of our intuition, by showing the superior performance with MHE regularization.

## 1 Introduction

The recent success of deep neural networks has led to its wide applications in a variety of tasks. With the over-parametrization nature and deep layered architecture, current deep networks [14, 46, 42] are able to achieve impressive performance on large-scale problems. Despite such success, having redundant and highly correlated neurons (*e.g.*, weights of kernels/filters in convolutional neural networks (CNNs)) caused by over-parametrization presents an issue [37, 41], which motivated a series of influential works in network compression [10, 1] and parameter-efficient network architectures [16, 19, 62]. These works either compress the network by pruning redundant neurons or directly modify the network architecture, aiming to achieve comparable performance while using fewer parameters. Yet, it remains an open problem to find a unified and principled theory that guides the network compression in the context of optimal generalization ability.

Another stream of works seeks to further release the network generalization power by alleviating redundancy through diversification [57, 56, 5, 36] as rigorously analyzed by [59]. Most of these works address the redundancy problem by enforcing relatively large diversity between pairwise projection bases via regularization. Our work broadly falls into this category by sharing similar high-level target, but the spirit and motivation behind our proposed models are distinct. In particular, there is a recent trend of studies that feature the significance of angular learning at both loss and convolution levels [29, 28, 30, 27], based on the observation that the angles in deep embeddings learned by CNNs tend to encode semantic difference. The key intuition is that angles preserve the most abundant and discriminative information for visual recognition. As a result, hyperspherical geodesic distances between neurons naturally play a key role in this context, and thus, it is intuitively desired to impose discrimination by keeping their projections on the hypersphere as far away from

each other as possible. While the concept of imposing large angular diversities was also considered in [59, 57, 56, 36], they do not consider diversity in terms of global equidistribution of embeddings on the hypersphere, which fails to achieve the state-of-the-art performances.

Given the above motivation, we draw inspiration from a well-known physics problem, called Thomson problem [48, 43]. The goal of Thomson problem is to determine the minimum electrostatic potential energy configuration of $N$ mutually-repelling electrons on the surface of a unit sphere. We identify the intrinsic resemblance between the Thomson problem and our target, in the sense that diversifying neurons can be seen as searching for an optimal configuration of electron locations. Similarly, we characterize the diversity for a group of neurons by defining a generic hyperspherical potential energy using their pairwise relationship. Higher energy implies higher redundancy, while lower energy indicates that these neurons are more diverse and more uniformly spaced. To reduce the redundancy of neurons and improve the neural networks, we propose a novel *minimum hyperspherical energy* (MHE) regularization framework, where the diversity of neurons is promoted by minimizing the hyperspherical energy in each layer. As verified by comprehensive experiments on multiple tasks, MHE is able to consistently improve the generalization power of neural networks.

MHE faces different situations when it is applied to hidden layers and output layers. For hidden layers, applying MHE straightforwardly may still encourage some degree of redundancy since it will produce co-linear bases pointing to opposite directions (see Fig. 1 middle). In order to avoid such redundancy, we propose the half-space MHE which constructs a group of virtual neurons and minimize the hyperspherical energy of both existing and virtual neurons. For output layers, MHE aims to distribute the classifier neurons[1] as uniformly as

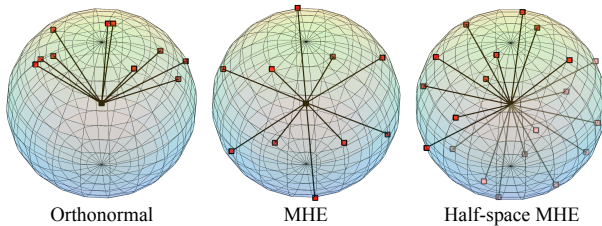

Figure 1: Orthonormal, MHE and half-space MHE regularization. The red dots denote the neurons optimized by the gradient of the corresponding regularization. The rightmost pink dots denote the virtual negative neurons. We randomly initialize the weights of 10 neurons on a 3D Sphere and optimize them with SGD.

possible to improve the inter-class feature separability. Different from MHE in hidden layers, classifier neurons should be distributed in the full space for the best classification performance [29, 28]. An intuitive comparison among the widely used orthonormal regularization, the proposed MHE and half-space MHE is provided in Fig. 1. One can observe that both MHE and half-space MHE are able to uniformly distribute the neurons over the hypersphere and half-space hypershpere, respectively. In contrast, conventional orthonormal regularization tends to group neurons closer, especially when the number of neurons is greater than the dimension.

MHE is originally defined on Euclidean distance, as indicated in Thomson problem. However, we further consider minimizing hyperspherical energy defined with respect to angular distance, which we will refer to as angular-MHE (A-MHE) in the following paper. In addition, we give some theoretical insights of MHE regularization, by discussing the asymptotic behavior and generalization error. Last, we apply MHE regularization to multiple vision tasks, including generic object recognition, class-imbalance learning, and face recognition. In the experiments, we show that MHE is architecture-agnostic and can considerably improve the generalization ability.

## 2 Related Works

Diversity regularization is shown useful in sparse coding [32, 35], ensemble learning [26, 24], self-paced learning [21], metric learning [58], etc. Early studies in sparse coding [32, 35] show that the generalization ability of codebook can be improved via diversity regularization, where the diversity is often modeled using the (empirical) covariance matrix. More recently, a series of studies have featured diversity regularization in neural networks [59, 57, 56, 5, 36, 55], where regularization is mostly achieved via promoting large angle/orthogonality, or reducing covariance between bases. Our work differs from these studies by formulating the diversity of neurons on the entire hypersphere, therefore promoting diversity from a more global, top-down perspective.

Methods other than diversity-promoting regularization have been widely proposed to improve CNNs [44, 20, 33, 30] and generative adversarial nets (GANs) [4, 34]. MHE can be regarded as a complement that can be applied on top of these methods.

# 3 Learning Neurons towards Minimum Hyperspherical Energy

## 3.1 Formulation of Minimum Hyperspherical Energy

Minimum hyperspherical energy defines an equilibrium state of the configuration of neuron's directions. We argue that the power of neural representation of each layer can be characterized by the hyperspherical energy of its neurons, and therefore a minimal energy configuration of neurons can induce better generalization. Before delving into details, we first define the hyperspherical energy functional for $N$ neurons (*i.e.*, kernels) with $(d+1)$-dimension $\boldsymbol{W}_N = \{\boldsymbol{w}_1, \cdots, \boldsymbol{w}_N \in \mathbb{R}^{d+1}\}$ as

$$\boldsymbol{E}_{s,d}(\hat{\boldsymbol{w}}_i|_{i=1}^N) = \sum_{i=1}^N \sum_{j=1, j \neq i}^N f_s\left(\|\hat{\boldsymbol{w}}_i - \hat{\boldsymbol{w}}_j\|\right) = \left\{ \begin{array}{ll} \sum_{i \neq j} \|\hat{\boldsymbol{w}}_i - \hat{\boldsymbol{w}}_j\|^{-s}, & s > 0 \\ \sum_{i \neq j} \log\left(\|\hat{\boldsymbol{w}}_i - \hat{\boldsymbol{w}}_j\|^{-1}\right), & s = 0 \end{array} \right., \quad (1)$$

where $\|\cdot\|$ denotes Euclidean distance, $f_s(\cdot)$ is a decreasing real-valued function, and $\hat{\boldsymbol{w}}_i = \frac{\boldsymbol{w}_i}{\|\boldsymbol{w}_i\|}$ is the $i$-th neuron weight projected onto the unit hypersphere $\mathbb{S}^d = \{\boldsymbol{w} \in \mathbb{R}^{d+1} | \|\boldsymbol{w}\| = 1\}$. We also denote $\hat{\boldsymbol{W}}_N = \{\hat{\boldsymbol{w}}_1, \cdots, \hat{\boldsymbol{w}}_N \in \mathbb{S}^d\}$, and $\boldsymbol{E}_s = \boldsymbol{E}_{s,d}(\hat{\boldsymbol{w}}_i|_{i=1}^N)$ for short. There are plenty of choices for $f_s(\cdot)$, but in this paper we use $f_s(z) = z^{-s}, s > 0$, known as Riesz $s$-kernels. Particularly, as $s \to 0$, $z^{-s} \to s \log(z^{-1}) + 1$, which is an affine transformation of $\log(z^{-1})$. It follows that optimizing the logarithmic hyperspherical energy $\boldsymbol{E}_0 = \sum_{i \neq j} \log(\|\hat{\boldsymbol{w}}_i - \hat{\boldsymbol{w}}_j\|^{-1})$ is essentially the limiting case of optimizing the hyperspherical energy $\boldsymbol{E}_s$. We therefore define $f_0(z) = \log(z^{-1})$ for convenience.

The goal of the MHE criterion is to minimize the energy in Eq. (1) by varying the orientations of the neuron weights $\boldsymbol{w}_1, \cdots, \boldsymbol{w}_N$. To be precise, we solve an optimization problem: $\min_{\boldsymbol{W}_N} \boldsymbol{E}_s$ with $s \geq 0$. In particular, when $s = 0$, we solve the logarithmic energy minimization problem:

$$\arg \min_{\boldsymbol{W}_N} \boldsymbol{E}_0 = \arg \min_{\boldsymbol{W}_N} \exp(\boldsymbol{E}_0) = \arg \max_{\boldsymbol{W}_N} \prod_{i \neq j} \|\hat{\boldsymbol{w}}_i - \hat{\boldsymbol{w}}_j\|, \quad (2)$$

in which we essentially maximize the product of Euclidean distances. $\boldsymbol{E}_0$, $\boldsymbol{E}_1$ and $\boldsymbol{E}_2$ have interesting yet profound connections. Note that Thomson problem corresponds to minimizing $\boldsymbol{E}_1$, which is a NP-hard problem. Therefore in practice we can only compute its approximate solution by heuristics. In neural networks, such a differentiable objective can be directly optimized via gradient descent.

## 3.2 Logarithmic Hyperspherical Energy $E_0$ as a Relaxation

Optimizing the original energy in Eq. (1) is equivalent to optimizing its logarithmic form $\log \boldsymbol{E}_s$. To efficiently solve this difficult optimization problem, we can instead optimize the lower bound of $\log \boldsymbol{E}_s$ as a surrogate energy, by applying Jensen's inequality:

$$\arg \min_{\boldsymbol{W}_N} \left\{ \boldsymbol{E}_{\log} := \sum_{i=1}^N \sum_{j=1, j \neq i}^N \log\left(f_s\left(\|\hat{\boldsymbol{w}}_i - \hat{\boldsymbol{w}}_j\|\right)\right) \right\} \quad (3)$$

With $f_s(z) = z^{-s}, s > 0$, we observe that $\boldsymbol{E}_{\log}$ becomes $s\boldsymbol{E}_0 = s \sum_{i \neq j} \log(\|\hat{\boldsymbol{w}}_i - \hat{\boldsymbol{w}}_j\|^{-1})$, which is identical to the logarithmic hyperspherical energy $\boldsymbol{E}_0$ up to a multiplicative factor $s$. Therefore, minimizing $\boldsymbol{E}_0$ can also be viewed as a relaxation of minimizing $\boldsymbol{E}_s$ for $s > 0$.

## 3.3 MHE as Regularization for Neural Networks

Now that we have introduced the formulation of MHE, we propose MHE regularization for neural networks. In supervised neural network learning, the entire objective function is shown as follows:

$$\mathcal{L} = \underbrace{\frac{1}{m} \sum_{j=1}^m \ell(\langle \boldsymbol{w}_i^{\text{out}}, \boldsymbol{x}_j \rangle_{i=1}^c, \boldsymbol{y}_j)}_{\text{training data fitting}} + \lambda_{\text{h}} \cdot \underbrace{\sum_{j=1}^{L-1} \frac{1}{N_j(N_j - 1)} \{\boldsymbol{E}_s\}_j}_{T_{\text{h}}: \text{ hyperspherical energy for hidden layers}} + \lambda_{\text{o}} \cdot \underbrace{\frac{1}{N_L(N_L - 1)} \boldsymbol{E}_s(\hat{\boldsymbol{w}}_i^{\text{out}}|_{i=1}^c)}_{T_{\text{o}}: \text{ hyperspherical energy for output layer}} \quad (4)$$

where $\boldsymbol{x}_i$ is the feature of the $i$-th training sample entering the output layer, $\boldsymbol{w}_i^{\text{out}}$ is the classifier neuron for the $i$-th class in the output fully-connected layer and $\hat{\boldsymbol{w}}_i^{\text{out}}$ denotes its normalized version. $\{\boldsymbol{E}_s\}_i$ denotes the hyperspherical energy for the neurons in the $i$-th layer. $c$ is the number of classes, $m$ is the batch size, $L$ is the number of layers of the neural network, and $N_i$ is the number of neurons in the $i$-th layer. $\boldsymbol{E}_s(\hat{\boldsymbol{w}}_i^{\text{out}}|_{i=1}^c)$ denotes the hyperspherical energy of neurons $\{\hat{\boldsymbol{w}}_1^{\text{out}}, \cdots, \hat{\boldsymbol{w}}_c^{\text{out}}\}$. The $\ell_2$ weight decay is omitted here for simplicity, but we will use it in practice. An alternative interpretation of MHE regularization from a decoupled view is given in Section 3.7 and Appendix C. MHE has different effects and interpretations in regularizing hidden layers and output layers.

**MHE for hidden layers.** To make neurons in the hidden layers more discriminative and less redundant, we propose to use MHE as a form of regularization. MHE encourages the normalized neurons to

be uniformly distributed on a unit hypersphere, which is partially inspired by the observation in [30] that angular difference in neurons preserves semantic (label-related) information. To some extent, MHE maximizes the average angular difference between neurons (specifically, the hyperspherical energy of neurons in every hidden layer). For instance, in CNNs we minimize the hyperpsherical energy of kernels in convolutional and fully-connected layers except the output layer.

**MHE for output layers.** For the output layer, we propose to enhance the inter-class feature separability with MHE to learn discriminative and well-separated features. For classification tasks, MHE regularization is complementary to the softmax cross-entropy loss in CNNs. The softmax loss focuses more on the intra-class compactness, while MHE encourages the inter-class separability. Therefore, MHE on output layers can induce features with better generalization power.

## 3.4  MHE in Half Space

Directly applying the MHE formulation may still encouter some redundancy. An example in Fig. 2, with two neurons in a 2-dimensional space, illustrates this potential issue. Directly imposing the original MHE regularization leads to a solution that two neurons are colinear but with opposite directions. To avoid such redundancy, we propose the half-space MHE regularization which constructs some virtual neurons and minimizes the hyperspherical energy of both original and virtual neurons together.

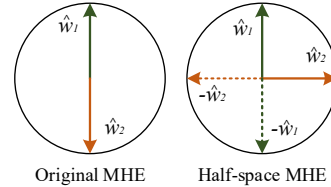

Figure 2: Half-space MHE.

Specifically, half-space MHE constructs a colinear virtual neuron with opposite direction for every existing neuron. Therefore, we end up with minimizing the hyperspherical energy with $2N_i$ neurons in the $i$-th layer (*i.e.*, minimizing $\boldsymbol{E}_s(\{\hat{\boldsymbol{w}}_k, -\hat{\boldsymbol{w}}_k\}|_{k=1}^{2N_i})$). This half-space variant will encourage the neurons to be less correlated and less redundant, as illustrated in Fig. 2. Note that, half-space MHE can only be used in hidden layers, because the colinear neurons do not constitute redundancy in output layers, as shown in [29]. Nevertheless, colinearity is usually not likely to happen in high-dimensional spaces, especially when the neurons are optimized to fit training data. This may be the reason that the original MHE regularization still consistently improves the baselines.

## 3.5  MHE beyond Euclidean Distance

The hyperspherical energy is originally defined based on the Euclidean distance on a hypersphere, which can be viewed as an angular measure. In addition to Euclidean distance, we further consider the geodesic distance on a unit hypersphere as a distance measure for neurons, which is exactly the same as the angle between neurons. Specifically, we consider to use $\arccos(\hat{\boldsymbol{w}}_i^\top \hat{\boldsymbol{w}}_j)$ to replace $\|\hat{\boldsymbol{w}}_i - \hat{\boldsymbol{w}}_j\|$ in hyperspherical energies. Following this idea, we propose angular MHE (A-MHE) as a simple extension, where the hyperspherical energy is rewritten as:

$$\boldsymbol{E}_{s,d}^a(\hat{\boldsymbol{w}}_i|_{i=1}^N) = \sum_{i=1}^N \sum_{j=1, j\neq i}^N f_s\big(\arccos(\hat{\boldsymbol{w}}_i^\top \hat{\boldsymbol{w}}_j)\big) = \begin{cases} \sum_{i\neq j} \arccos(\hat{\boldsymbol{w}}_i^\top \hat{\boldsymbol{w}}_j)^{-s}, & s > 0 \\ \sum_{i\neq j} \log\big(\arccos(\hat{\boldsymbol{w}}_i^\top \hat{\boldsymbol{w}}_j)^{-1}\big), & s = 0 \end{cases} \quad (5)$$

which can be viewed as redefining MHE based on geodesic distance on hyperspheres (*i.e.*, angle), and can be used as an alternative to the original hyperspherical energy $\boldsymbol{E}_s$ in Eq. (4). Note that, A-MHE can also be learned in full-space or half-space, leading to similar variants as original MHE. The key difference between MHE and A-MHE lies in the optimization dynamics, because their gradients w.r.t the neuron weights are quite different. A-MHE is also more computationally expensive than MHE.

## 3.6  Mini-batch Approximation for MHE

With a large number of neurons in one layer, calculating MHE can be computationally expensive as it requires computing the pair-wise distances between neurons. To address this issue, we propose the mini-batch version of MHE to approximate the MHE (either original or half-space) objective.

**Mini-batch approximation for MHE on hidden layers.** For hidden layers, mini-batch approximation iteratively takes a random batch of neurons as input and minimizes their hyperspherical energy as an approximation to the MHE. Note that the gradient of the mini-batch objective is an unbiased estimation of the original gradient of MHE.

**Data-dependent mini-batch approximation for output layers.** For the output layer, the data-dependent mini-batch approximation iteratively takes the classifier neurons corresponding to the classes that exist in mini-batches. It minimizes $\frac{1}{m(N-1)} \sum_{i=1}^m \sum_{j=1, j\neq y_i}^N f_s(\|\hat{\boldsymbol{w}}_{y_i} - \hat{\boldsymbol{w}}_j\|)$ in each iteration, where $y_i$ denotes the class label of the $i$-th sample in each mini-batch, $m$ is the mini-batch size, and $N$ is the number of neurons (in one particular layer).

### 3.7 Discussions

**Connections to scientific problems.** The hyperspherical energy minimization has close relationships with scientific problems. When $s = 1$, Eq. (1) reduces to Thomson problem [48, 43] (in physics) where one needs to determine the minimum electrostatic potential energy configuration of $N$ mutually-repelling electrons on a unit sphere. When $s = \infty$, Eq. (1) becomes Tammes problem [47] (in geometry) where the goal is to pack a given number of circles on the surface of a sphere such that the minimum distance between circles is maximized. When $s = 0$, Eq. (1) becomes Whyte's problem where the goal is to maximize product of Euclidean distances as shown in Eq. (2). Our work aims to make use of important insights from these scientific problems to improve neural networks.

**Understanding MHE from decoupled view.** Inspired by decoupled networks [27], we can view the original convolution as the multiplication of the angular function $g(\theta) = \cos(\theta)$ and the magnitude function $h(\|\boldsymbol{w}\|, \|\boldsymbol{x}\|) = \|\boldsymbol{w}\| \cdot \|\boldsymbol{x}\|$: $f(\boldsymbol{w}, \boldsymbol{x}) = h(\|\boldsymbol{w}\|, \|\boldsymbol{x}\|) \cdot g(\theta)$ where $\theta$ is the angle between the kernel $\boldsymbol{w}$ and the input $\boldsymbol{x}$. From the equation above, we can see that the norm of the kernel and the direction (*i.e.*, angle) of the kernel affect the inner product similarity differently. Typically, weight decay is to regularize the kernel by minimizing its $\ell_2$ norm, while there is no regularization on the direction of the kernel. Therefore, MHE completes this missing piece by promoting angular diversity. By combining MHE to a standard neural networks, the entire regularization term becomes

$$\mathcal{L}_{\text{reg}} = \underbrace{\lambda_{\text{w}} \cdot \frac{1}{\sum_{j=1}^{L} N_j} \sum_{j=1}^{L} \sum_{i=1}^{N_j} \|\boldsymbol{w}_i\|}_{\text{Weight decay: regularizing the magnitude of kernels}} + \underbrace{\lambda_{\text{h}} \cdot \sum_{j=1}^{L-1} \frac{1}{N_j(N_j-1)} \{\boldsymbol{E}_s\}_j + \lambda_{\text{o}} \cdot \frac{1}{N_L(N_L-1)} \boldsymbol{E}_s(\hat{\boldsymbol{w}}_i^{\text{out}}|_{i=1}^{c})}_{\text{MHE: regularizing the direction of kernels}}$$

where $\lambda_{\text{w}}$, $\lambda_{\text{h}}$ and $\lambda_{\text{o}}$ are weighting hyperparameters for these three regularization terms. From the decoupled view, MHE makes a lot of senses in regularizing the neural networks, since it serves as a complementary and orthogonal role to weight decay. More discussions are in Appendix C.

**Comparison to orthogonality/angle-promoting regularizations.** Promoting orthogonality or large angles between bases has been a popular choice for encouraging diversity. Probably the most related and widely used one is the orthonormal regularization [30] which aims to minimize $\|\boldsymbol{W}^\top \boldsymbol{W} - \boldsymbol{I}\|_F$, where $W$ denotes the weights of a group of neurons with each column being one neuron and $\boldsymbol{I}$ is an identity matrix. One similar regularization is the orthogonality regularization [36] which minimizes the sum of the cosine values between all the kernel weights. These methods encourage kernels to be orthogonal to each other, while MHE does not. Instead, MHE encourages the hyperspherical diversity among these kernels, and these kernels are not necessarily orthogonal to each other. [56] proposes the angular constraint which aims to constrain the angles between different kernels of the neural network, but quite different from MHE, they use a hard constraint to impose this angular regularization. Moreover, these methods model diversity regularization at a more local level, while MHE regularization seeks to model the problem in a more top-down manner.

**Normalized neurons in MHE.** From Eq. 1, one can see that the normalized neurons are used to compute MHE, because we aim to encourage the diversity on a hypersphere. However, a natural question may arise: what if we use the original (*i.e.*, unnormalized) neurons to compute MHE? First, combining the norm of kernels (*i.e.*, neurons) into MHE may lead to a trivial gradient descent direction: simply increasing the norm of all kernels. Suppose all kernel directions stay unchanged, increasing the norm of all kernels by a factor can effectively decrease the objective value of MHE. Second, coupling the norm of kernels into MHE may contradict with weight decay which aims to decrease the norm of kernels. Moreover, normalized neurons imply that the importance of all neurons is the same, which matches the intuition in [28, 30, 27]. If we desire different importance for different neurons, we can also manually assign a fixed weight for each neuron. This may be useful when we have already known certain neurons are more important and we want them to be relatively fixed. The neuron with large weight tends to be updated less. We will discuss it more in Appendix D.

## 4 Theoretical Insights

This section leverages a number of rigorous theoretical results from [38, 23, 12, 25, 11, 23, 8, 54] and provides theoretical yet intuitive understandings about MHE.

### 4.1 Asymptotic Behavior

This subsection shows how the hyperspherical energy behaves asymptotically. Specifically, as $N \to \infty$, we can show that the solution $\hat{\boldsymbol{W}}_N$ tends to be uniformly distributed on hypersphere $\mathbb{S}^d$ when the hyperspherical energy defined in Eq. (1) achieves its minimum.

**Definition 1** (minimal hyperspherical $s$-energy)**.** *We define the minimal $s$-energy for $N$ points on the unit hypersphere $\mathbb{S}^d = \{ w \in \mathbb{R}^{d+1} | \, \|\boldsymbol{w}\| = 1 \}$ as*

$$\boldsymbol{\varepsilon}_{s,d}(N) \coloneqq \inf_{\hat{\boldsymbol{W}}_N \subset \mathbb{S}^d} \boldsymbol{E}_{s,d}(\hat{\boldsymbol{w}}_i|_{i=1}^N) \tag{6}$$

*where the infimum is taken over all possible $\hat{\boldsymbol{W}}_N$ on $\mathbb{S}^d$. Any configuration of $\hat{\boldsymbol{W}}_N$ to attain the infimum is called an $s$-extremal configuration. Usually $\boldsymbol{\varepsilon}_{s,d}(N) = \infty$ if $N$ is greater than $d$ and $\boldsymbol{\varepsilon}_{s,d}(N) = 0$ if $N = 0, 1$.*

We discuss the asymptotic behavior ($N \to \infty$) in three cases: $0 < s < d$, $s = d$, and $s > d$. We first write the energy integral as $I_s(\mu) = \iint_{\mathbb{S}^d \times \mathbb{S}^d} \|\boldsymbol{u} - \boldsymbol{v}\|^{-s} d\mu(\boldsymbol{u}) d\mu(\boldsymbol{v})$, which is taken over all probability measure $\mu$ supported on $\mathbb{S}^d$. With $0 < s < d$, $I_s(\mu)$ is minimal when $\mu$ is the spherical measure $\sigma^d = \mathcal{H}^d(\cdot)|_{\mathbb{S}^d} / \mathcal{H}^d(\mathbb{S}^d)$ on $\mathbb{S}^d$, where $\mathcal{H}^d(\cdot)$ denotes the $d$-dimensional Hausdorff measure. When $s \geq d$, $I_s(\mu)$ becomes infinity, which therefore requires different analysis. In general, we can say all $s$-extremal configurations asymptotically converge to uniform distribution on a hypersphere, as stated in Theorem 1. This asymptotic behavior has been heavily studied in [38, 23, 12].

**Theorem 1** (asymptotic uniform distribution on hypersphere)**.** *Any sequence of optimal $s$-energy configurations $(\hat{\boldsymbol{W}}_N^\star)|_2^\infty \subset \mathbb{S}^d$ is asymptotically uniformly distributed on $\mathbb{S}^d$ in the sense of the weak-star topology of measures, namely*

$$\frac{1}{N} \sum_{\boldsymbol{v} \in \hat{\boldsymbol{W}}_N^\star} \delta_{\boldsymbol{v}} \to \sigma^d, \quad \text{as } N \to \infty \tag{7}$$

*where $\delta_{\boldsymbol{v}}$ denotes the unit point mass at $\boldsymbol{v}$, and $\sigma^d$ is the spherical measure on $\mathbb{S}^d$.*

**Theorem 2** (asymptotics of the minimal hyperspherical $s$-energy)**.** *We have that $\lim_{N \to \infty} \frac{\boldsymbol{\varepsilon}_{s,d}(N)}{p(N)}$ exists for the minimal $s$-energy. For $0 < s < d$, $p(N) = N^2$. For $s = d$, $p(N) = N^2 \log N$. For $s > d$, $p(N) = N^{1+s/d}$. Particularly if $0 < s < d$, we have $\lim_{N \to \infty} \frac{\boldsymbol{\varepsilon}_{s,d}(N)}{N^2} = I_s(\sigma^d)$.*

Theorem 2 tells us the growth power of the minimal hyperspherical $s$-energy when $N$ goes to infinity. Therefore, different potential power $s$ leads to different optimization dynamics. In the light of the behavior of the energy integral, MHE regularization will focus more on local influence from neighborhood neurons instead of global influences from all the neurons as the power $s$ increases.

### 4.2 Generalization and Optimality

As proved in [54], in one-hidden-layer neural network, the diversity of neurons can effectively eliminate the spurious local minima despite the non-convexity in learning dynamics of neural networks. Following such an argument, our MHE regularization, which encourages the diversity of neurons, naturally matches the theoretical intuition in [54], and effectively promotes the generalization of neural networks. While hyperspherical energy is minimized such that neurons become diverse on hyperspheres, the hyperspherical diversity is closely related to the generalization error.

More specifically, in a one-hidden-layer neural network $f(x) = \sum_{k=1}^n v_k \sigma(\boldsymbol{W}_k^\top \boldsymbol{x})$ with least squares loss $L(f) = \frac{1}{2m} \sum_{i=1}^m (y_i - f(\boldsymbol{x}_i))^2$, we can compute its gradient w.r.t $\boldsymbol{W}_k$ as $\frac{\partial L}{\partial \boldsymbol{W}_k} = \frac{1}{m} \sum_{i=1}^m (f(x_i) - y_i) v_k \sigma'(\boldsymbol{W}_k^\top \boldsymbol{x}_i) \boldsymbol{x}_i$. ($\sigma(\cdot)$ is the nonlinear activation function and $\sigma'(\cdot)$ is its subgradient. $\boldsymbol{x} \in$ is the training sample. $\boldsymbol{W}_k$ denotes the weights of hidden layer and $v_k$ is the weights of output layer.) Subsequently, we can rewrite this gradient as a matrix form: $\frac{\partial L}{\partial \boldsymbol{W}} = \boldsymbol{D} \cdot \boldsymbol{r}$ where $\boldsymbol{D} \in \mathbb{R}^{dn \times m}$, $\boldsymbol{D}_{\{di-d+1:di, j\}} = v_i \sigma'(\boldsymbol{W}_i^\top \boldsymbol{x}_j) \boldsymbol{x}_j \in \mathbb{R}^d$ and $\boldsymbol{r} \in \mathbb{R}^m$, $\boldsymbol{r}_i = \frac{1}{m} f(\boldsymbol{x}_i) - y_i$. Further, we can obtain the inequality $\|\boldsymbol{r}\| \leq \frac{1}{\lambda_{\min}(\boldsymbol{D})} \|\frac{\partial L}{\partial \boldsymbol{W}}\|$. $\|\boldsymbol{r}\|$ is actually the training error. To make the training error small, we need to lower bound $\lambda_{\min}(\boldsymbol{D})$ away from zero. From [54, 3], one can know that the lower bound of $\lambda_{\min}(\boldsymbol{D})$ is directly related to the hyperspherical diversity of neurons. After bounding the training error, it is easy to bound the generalization error using Rademachar complexity.

## 5 Applications and Experiments

### 5.1 Improving Network Generalization

First, we perform ablation study and some exploratory experiments on MHE. Then we apply MHE to large-scale object recognition and class-imbalance learning. For all the experiments on CIFAR-10 and CIFAR-100 in the paper, we use moderate data augmentation, following [14, 27]. For ImageNet-2012, we follow the same data augmentation in [30]. We train all the networks using SGD with momentum 0.9, and the network initialization follows [13]. All the networks use BN [20] and ReLU if not otherwise specified. Experimental details are given in each subsection and Appendix A.

### 5.1.1 Ablation Study and Exploratory Experiments

**Variants of MHE.** We evaluate all different variants of MHE on CIFAR-10 and CIFAR-100, including original MHE (with the power $s = 0, 1, 2$) and half-space MHE (with the power $s = 0, 1, 2$) with both Euclidean and angular distance. In this experiment, all methods use CNN-9

| Method | CIFAR-10 | | | CIFAR-100 | | |
|---|---|---|---|---|---|---|
| | $s=2$ | $s=1$ | $s=0$ | $s=2$ | $s=1$ | $s=0$ |
| MHE | 6.22 | 6.74 | 6.44 | 27.15 | 27.09 | **26.16** |
| Half-space MHE | 6.28 | 6.54 | **6.30** | **25.61** | **26.30** | 26.18 |
| A-MHE | **6.21** | 6.77 | 6.45 | 26.17 | 27.31 | 27.90 |
| Half-space A-MHE | 6.52 | **6.49** | 6.44 | 26.03 | 26.52 | 26.47 |
| Baseline | 7.75 | | | 28.13 | | |

Table 1: Testing error (%) of different MHE on CIFAR-10/100.

(see Appendix A). The results in Table 1 show that all the variants of MHE perform consistently better than the baseline. Specifically, the half-space MHE has more significant performance gain compared to the other MHE variants, and MHE with Euclidean and angular distance perform similarly. In general, MHE with $s = 2$ performs best among $s = 0, 1, 2$. In the following experiments, we use $s = 2$ and Euclidean distance for both MHE and half-space MHE by default if not otherwise specified.

**Network width.** We evaluate MHE with different network width. We use CNN-9 as our base network, and change its filter number in Conv1.x, Conv2.x and Conv3.x (see Appendix A) to 16/32/64, 32/64/128,

| Method | 16/32/64 | 32/64/128 | 64/128/256 | 128/256/512 | 256/512/1024 |
|---|---|---|---|---|---|
| Baseline | 47.72 | 38.64 | 28.13 | 24.95 | 25.45 |
| MHE | 36.84 | 30.05 | 26.75 | 24.05 | 23.14 |
| Half-space MHE | **35.16** | **29.33** | **25.96** | **23.38** | **21.83** |

Table 2: Testing error (%) of different width on CIFAR-100.

64/128/256, 128/256/512 and 256/512/1024. Results in Table 2 show that both MHE and half-space MHE consistently outperform the baseline, showing stronger generalization. Interestingly, both MHE and half-space MHE have more significant gain while the filter number is smaller in each layer, indicating that MHE can help the network to make better use of the neurons. In general, half-space MHE performs consistently better than MHE, showing the necessity of reducing colinearity redundancy among neurons. Both MHE and half-space MHE outperform the baseline with a huge margin while the network is either very wide or very narrow, showing the superiority in improving generalization.

**Network depth.** We perform experiments with different network depth to better evaluate the performance of MHE. We fix the filter number in Conv1.x, Conv2.x and Conv3.x to 64, 128 and 256, respectively. We compare 6-layer CNN, 9-layer CNN and 15-layer CNN. The results are given in Table 3. Both MHE and half-space MHE perform significantly better

| Method | CNN-6 | CNN-9 | CNN-15 |
|---|---|---|---|
| Baseline | 32.08 | 28.13 | N/C |
| MHE | 28.16 | 26.75 | 26.9 |
| Half-space MHE | **27.56** | **25.96** | **25.84** |

Table 3: Testing error (%) of different depth on CIFAR-100. N/C: not converged.

than the baseline. More interestingly, baseline CNN-15 can not converge, while CNN-15 is able to converge reasonably well if we use MHE to regularize the network. Moreover, we also see that half-space MHE can consistently show better generalization than MHE with different network depth.

**Ablation study.** Since the current MHE regularizes the neurons in the hidden layers and the output layer simultaneously, we perform ablation study for MHE to further investigate where the gain comes from. This experiment uses the CNN-9. The results are given in Table 4. "H" means that we apply MHE to all the hidden layers, while "O" means that we apply MHE to the output layer. Because the half-space MHE can not be

| Method | H O<br>$\times\ \checkmark$ | H O<br>$\checkmark\ \times$ | H O<br>$\checkmark\ \checkmark$ |
|---|---|---|---|
| MHE | 26.85 | 26.55 | **26.16** |
| Half-space MHE | N/A | 26.28 | **25.61** |
| A-MHE | 27.8 | 26.56 | **26.17** |
| Half-space A-MHE | N/A | 26.64 | **26.03** |
| Baseline | 28.13 | | |

Table 4: Ablation study on CIFAR-100.

applied to the output layer, so there is "N/A" in the table. In general, we find that applying MHE to both the hidden layers and the output layer yields the best performance, and using MHE in the hidden layers usually produces better accuracy than using MHE in the output layer.

**Hyperparameter experiment.** We evaluate how the selection of hyperparameter affects the performance. We experiment with different hyperparameters from $10^{-2}$ to $10^2$ on CIFAR-100 with the CNN-9. HS-MHE denotes the half-space MHE. We evaluate MHE variants by separately applying MHE to the output layer ("O"), MHE to the hidden layers ("H"), and the half-space MHE to the hidden layers ("H"). The results in Fig. 3 show that our MHE is not very hyperparameter-sensitive and can consistently beat the baseline by a considerable margin. One can observe that MHE's hyperparameter works well from $10^{-2}$ to $10^2$ and therefore is easy to set. In contrast, the hyperparameter of weight decay could be more sensitive than MHE. Half-space MHE can consistently

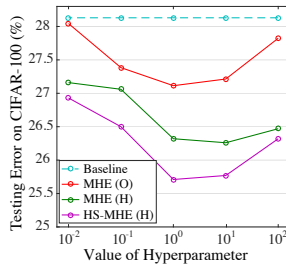

Figure 3: Hyperparameter.

outperform the original MHE under all different hyperparameter settings. Interestingly, applying MHE only to hidden layers can achieve better accuracy than applying MHE only to output layers.

**MHE for ResNets.** Besides the standard CNN, we also evaluate MHE on ResNet-32 to show that our MHE is architecture-agnostic and can improve accuracy on multiple types of architectures. Besides ResNets, MHE can also be applied to GoogleNet [46], SphereNets [30] (the experimental results are given in Appendix E), DenseNet [17], etc. Detailed architecture settings are given in Appendix A.

| Method | CIFAR-10 | CIFAR-100 |
|---|---|---|
| ResNet-110-original [14] | 6.61 | 25.16 |
| ResNet-1001 [15] | 4.92 | **22.71** |
| ResNet-1001 (64 batch) [15] | **4.64** | - |
| baseline | 5.19 | 22.87 |
| MHE | 4.72 | 22.19 |
| Half-space MHE | **4.66** | **22.04** |

Table 5: Error (%) of ResNet-32.

The results on CIFAR-10 and CIFAR-100 are given in Table 5. One can observe that applying MHE to ResNet also achieves considerable improvements, showing that MHE is generally useful for different architectures. Most importantly, adding MHE regularization will not affect the original architecture settings, and it can readily improve the network generalization at a neglectable computational cost.

### 5.1.2 Large-scale Object Recognition

We evaluate MHE on large-scale ImageNet-2012 datasets. Specifically, we perform experiment using ResNets, and then report the top-1 validation error (center crop) in Table 6. From the results, we still observe that both MHE and half-space MHE yield consistently better recognition accuracy than the baseline and the orthonormal regularization (after tuning its hyperparameter). To

| Method | ResNet-18 | ResNet-34 |
|---|---|---|
| baseline | 33.95 | 30.04 |
| Orthogonal [36] | 33.65 | 29.74 |
| Orthnormal | 33.61 | 29.75 |
| MHE | 33.50 | 29.60 |
| Half-space MHE | **33.45** | **29.50** |

Table 6: Top1 error (%) on ImageNet.

better evaluate the consistency of MHE's performance gain, we use two ResNets with different depth: ResNet-18 and ResNet-34. On these two different networks, both MHE and half-space MHE outperform the baseline by a significant margin, showing consistently better generalization power. Moreover, half-space MHE performs slightly better than full-space MHE as expected.

### 5.1.3 Class-imbalance Learning

Because MHE aims to maximize the hyperspherical margin between different classifier neurons in the output layer, we can naturally apply MHE to class-imbalance learning where the number of training samples in different classes is imbalanced. We demonstrate the power of MHE in class-imbalance learning through a toy experiment. We first randomly throw away 98% training data for digit 0 in MNIST (only 100 samples are preserved for digit 0), and then train a 6-layer CNN on this imbalance MNIST. To visualize the learned features, we set

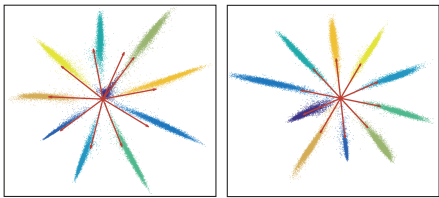

(a) CNN without MHE     (b) CNN with MHE

Figure 4: Class-imbalance learning on MNIST.

the output feature dimension as 2. The features and classifier neurons on the full training set are visualized in Fig. 4 where each color denotes a digit and red arrows are the normalized classifier neurons. Although we train the network on the imbalanced training set, we visualize the features of the full training set for better demonstration. The visualization for the full testing set is also given in Appendix H. From Fig. 4, one can see that the CNN without MHE tends to ignore the imbalanced class (digit 0) and the learned classifier neuron is highly biased to another digit. In contrast, the CNN with MHE can learn reasonably separable distribution even if digit 0 only has 2% samples compared to the other classes. Using MHE in this toy setting can readily improve the accuracy on the full testing set from 88.5% to 98%. Most importantly, the classifier neuron for digit 0 is also properly learned, similar to the one learned on the balanced dataset. Note that, half-space MHE can not be applied to the classifier neurons, because the classifier neurons usually need to occupy the full feature space.

We experiment MHE in two data imbalance settings on CIFAR-10: 1) single class imbalance (S) - All classes have the same number of images but one single class has significantly less number, and 2) multiple class imbalance (M) - The number of images decreases as the class index decreases from 9 to 0. We use CNN-9 for all the compared regularizations. Detailed setups are provided in Appendix A. In

| Method | Single | Err. (S) | Multiple |
|---|---|---|---|
| Baseline | 9.80 | 30.40 | 12.00 |
| Orthonormal | 8.34 | 26.80 | 10.80 |
| MHE | 7.98 | **25.80** | 10.25 |
| Half-space MHE | 7.90 | 26.40 | **9.59** |
| A-MHE | 7.96 | 26.00 | 9.88 |
| Half-space A-MHE | **7.59** | 25.90 | 9.89 |

Table 7: Error on imbalanced CIFAR-10.

Table 7, we report the error rate on the whole testing set. In addition, we report the error rate (denoted by Err. (S)) on the imbalance class (single imbalance setting) in the full testing set. From the results, one can observe that CNN-9 with MHE is able to effectively perform recognition when classes are imbalanced. Even only given a small portion of training data in a few classes, CNN-9 with MHE can achieve very competitive accuracy on the full testing set, showing MHE's superior generalization power. Moreover, we also provide experimental results on imbalanced CIFAR-100 in Appendix H.

## 5.2 *SphereFace+*: Improving Inter-class Feature Separability via MHE for Face Recognition

We have shown that full-space MHE for output layers can encourage classifier neurons to distribute more evenly on hypersphere and therefore improve inter-class feature separability. Intuitively, the classifier neurons serve as the approximate center for features from each class, and can therefore guide the feature learning. We also observe that open-set face recognition (*e.g.*, face verification) requires the feature centers to be as separable as possible [28]. This connection inspires us to apply MHE to face recognition. Specifically, we propose *SphereFace+* by applying MHE to SphereFace [28]. The objective of SphereFace, angular softmax loss ($\ell_{\text{SF}}$) that encourages intra-class feature compactness, is naturally complementary to that of MHE. The objective function of SphereFace+ is defined as

$$\mathcal{L}_{\text{SF+}} = \underbrace{\frac{1}{m}\sum_{j=1}^{m}\ell_{\text{SF}}(\langle \boldsymbol{w}_i^{\text{out}}, \boldsymbol{x}_j \rangle_{i=1}^{c}, \boldsymbol{y}_j, m_{\text{SF}})}_{\text{Angular softmax loss: promoting } \textbf{intra-class compactness}} + \underbrace{\lambda_{\text{M}} \cdot \frac{1}{m(N-1)}\sum_{i=1}^{m}\sum_{j=1, j\neq y_i}^{N} f_s(\|\hat{\boldsymbol{w}}_{y_i}^{\text{out}} - \hat{\boldsymbol{w}}_j^{\text{out}}\|)}_{\text{MHE: promoting } \textbf{inter-class separability}} \quad (8)$$

where $c$ is the number of classes, $m$ is the mini-batch size, $N$ is the number of classifier neurons, $\boldsymbol{x}_i$ the deep feature of the $i$-th face ($y_i$ is its groundtruth label), and $\boldsymbol{w}_i^{\text{out}}$ is the $i$-th classifier neuron. $m_{\text{SF}}$ is a hyperparameter for SphereFace, controlling the degree of intra-class feature compactness (*i.e.*, the size of the angular margin). Because face datesets usually have thousands of identities, we will use the data-dependent mini-batch approximation MHE as shown in Eq. (8) in the output layer to reduce computational cost. MHE completes a missing piece for SphereFace by promoting the inter-class separability. SphereFace+ consistently outperforms SphereFace, and achieves state-of-the-art performance on both LFW [18] and MegaFace [22] datasets. More results on MegaFace are put in Appendix I. MHE can also improve other face recognition methods, as shown in Appendix F.

| $m_{\text{SF}}$ | LFW | | MegaFace | |
|---|---|---|---|---|
| | SphereFace | SphereFace+ | SphereFace | SphereFace+ |
| 1 | 96.35 | **97.15** | 39.12 | **45.90** |
| 2 | 98.87 | **99.05** | 60.48 | **68.51** |
| 3 | 98.97 | **99.13** | 63.71 | **66.89** |
| 4 | 99.26 | **99.32** | 70.68 | **71.30** |

Table 8: Accuracy (%) on SphereFace-20 network.

| $m_{\text{SF}}$ | LFW | | MegaFace | |
|---|---|---|---|---|
| | SphereFace | SphereFace+ | SphereFace | SphereFace+ |
| 1 | 96.93 | **97.47** | 41.07 | **45.55** |
| 2 | 99.03 | **99.22** | 62.01 | **67.07** |
| 3 | 99.25 | **99.35** | 69.69 | **70.89** |
| 4 | 99.42 | **99.47** | 72.72 | **73.03** |

Table 9: Accuracy (%) on SphereFace-64 network.

**Performance under different $m_{\text{SF}}$.** We evaluate SphereFace+ with two different architectures (SphereFace-20 and SphereFace-64) proposed in [28]. Specifically, SphereFace-20 and SphereFace-64 are 20-layer and 64-layer modified residual networks, respectively. We train our network with the publicly available CASIA-Webface dataset [60], and then test the learned model on LFW and MegaFace dataset. In MegaFace dataset, the reported accuracy indicates rank-1 identification accuracy with 1 million distractors. All the results in Table 8 and Table 9 are computed without model ensemble and PCA. One can observe that SphereFace+ consistently outperforms SphereFace by a considerable margin on both LFW and MegaFace datasets under all different settings of $m_{\text{SF}}$. Moreover, the performance gain generalizes across network architectures with different depth.

**Comparison to state-of-the-art methods.** We also compare our methods with some widely used loss functions. All these compared methods use SphereFace-64 network that are trained with CASIA dataset. All the results are given in Table 10 computed without model ensemble and PCA. Compared to the other state-of-the-art methods, SphereFace+ achieves the best accuracy on LFW dataset, while being comparable to the best accuracy on MegaFace dataset. Current state-of-the-art face recognition methods [49, 28, 51, 6, 31] usually only focus on compressing the intra-class features, which makes MHE a potentially useful tool in order to further improve these face recognition methods.

| Method | LFW | MegaFace |
|---|---|---|
| Softmax Loss | 97.88 | 54.86 |
| Softmax+Contrastive [45] | 98.78 | 65.22 |
| Triplet Loss [40] | 98.70 | 64.80 |
| L-Softmax Loss [29] | 99.10 | 67.13 |
| Softmax+Center Loss [53] | 99.05 | 65.49 |
| CosineFace [51, 49] | **99.10** | **75.10** |
| SphereFace | 99.42 | 72.72 |
| SphereFace+ (ours) | **99.47** | **73.03** |

Table 10: Comparison to state-of-the-art.

## 6 Concluding Remarks

We borrow some useful ideas and insights from physics and propose a novel regularization method for neural networks, called minimum hyperspherical energy (MHE), to encourage the angular diversity of neuron weights. MHE can be easily applied to every layer of a neural network as a plug-in regularization, without modifying the original network architecture. Different from existing methods, such diversity can be viewed as uniform distribution over a hypersphere. In this paper, MHE has been specifically used to improve network generalization for generic image classification, class-imbalance learning and large-scale face recognition, showing consistent improvements in all tasks. Moreover, MHE can significantly improve the image generation quality of GANs (see Appendix G). In summary, our paper casts a novel view on regularizing the neurons by introducing hyperspherical diversity.

## Acknowledgements

This project was supported in part by NSF IIS-1218749, NIH BIGDATA 1R01GM108341, NSF CAREER IIS-1350983, NSF IIS-1639792 EAGER, NSF IIS-1841351 EAGER, NSF CCF-1836822, NSF CNS-1704701, ONR N00014-15-1-2340, Intel ISTC, NVIDIA, Amazon AWS and Siemens. We would like to thank NVIDIA corporation for donating Titan Xp GPUs to support our research. We also thank Tuo Zhao for the valuable discussions and suggestions.

## Footnotes

* indicates equal contributions. Correspondence to: Weiyang Liu <wyliu@gatech.edu>.

[1]Classifier neurons are the projection bases of the last layer (*i.e.*, output layer) before input to softmax.

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
