[Supplementary Material]

# Appendix

## A  Experimental Details

| Layer | CNN-6 | CNN-9 | CNN-15 |
|---|---|---|---|
| Conv1.x | [3×3, 64]×2 | [3×3, 64]×3 | [3×3, 64]×5 |
| Pool1 | 2×2 Max Pooling, Stride 2 | | |
| Conv2.x | [3×3, 128]×2 | [3×3, 128]×3 | [3×3, 128]×5 |
| Pool2 | 2×2 Max Pooling, Stride 2 | | |
| Conv3.x | [3×3, 256]×2 | [3×3, 256]×3 | [3×3, 256]×5 |
| Pool3 | 2×2 Max Pooling, Stride 2 | | |
| Fully Connected | 256 | 256 | 256 |

Table 11: Our plain CNN architectures with different convolutional layers. Conv1.x, Conv2.x and Conv3.x denote convolution units that may contain multiple convolution layers. E.g., [3×3, 64]×3 denotes 3 cascaded convolution layers with 64 filters of size 3×3.

| Layer | ResNet-32 for CIFAR-10/100 | ResNet-18 for ImageNet-2012 | ResNet-34 for ImageNet-2012 |
|---|---|---|---|
| Conv0.x | N/A | [7×7, 64], Stride 2 <br> 3×3, Max Pooling, Stride 2 | [7×7, 64], Stride 2 <br> 3×3, Max Pooling, Stride 2 |
| Conv1.x | [3×3, 64]×1 <br> $\begin{bmatrix} 3 \times 3, 64 \\ 3 \times 3, 64 \end{bmatrix} \times 5$ | $\begin{bmatrix} 3 \times 3, 64 \\ 3 \times 3, 64 \end{bmatrix} \times 2$ | $\begin{bmatrix} 3 \times 3, 64 \\ 3 \times 3, 64 \end{bmatrix} \times 3$ |
| Conv2.x | $\begin{bmatrix} 3 \times 3, 128 \\ 3 \times 3, 128 \end{bmatrix} \times 5$ | $\begin{bmatrix} 3 \times 3, 128 \\ 3 \times 3, 128 \end{bmatrix} \times 2$ | $\begin{bmatrix} 3 \times 3, 128 \\ 3 \times 3, 128 \end{bmatrix} \times 4$ |
| Conv3.x | $\begin{bmatrix} 3 \times 3, 256 \\ 3 \times 3, 256 \end{bmatrix} \times 5$ | $\begin{bmatrix} 3 \times 3, 256 \\ 3 \times 3, 256 \end{bmatrix} \times 2$ | $\begin{bmatrix} 3 \times 3, 256 \\ 3 \times 3, 256 \end{bmatrix} \times 6$ |
| Conv4.x | N/A | $\begin{bmatrix} 3 \times 3, 512 \\ 3 \times 3, 512 \end{bmatrix} \times 2$ | $\begin{bmatrix} 3 \times 3, 512 \\ 3 \times 3, 512 \end{bmatrix} \times 3$ |
| | Average Pooling | | |

Table 12: Our ResNet architectures with different convolutional layers. Conv0.x, Conv1.x, Conv2.x, Conv3.x and Conv4.x denote convolution units that may contain multiple convolutional layers, and residual units are shown in double-column brackets. Conv1.x, Conv2.x and Conv3.x usually operate on different size feature maps. These networks are essentially the same as [14], but some may have a different number of filters in each layer. The downsampling is performed by convolutions with a stride of 2. E.g., [3×3, 64]×4 denotes 4 cascaded convolution layers with 64 filters of size 3×3, and S2 denotes stride 2.

**General settings.** The network architectures used in the paper are elaborated in Table 11 Table 12. For CIFAR-10 and CIFAR-100, we use batch size 128. We start with learning rate 0.1, divide it by 10 at 20k, 30k and 37.5k iterations, and terminate training at 42.5k iterations. For ImageNet-2012, we use batch size 64 and start with learning rate 0.1. The learning rate is divided by 10 at 150k, 300k and 400k iterations, and the training is terminated at 500k iterations. Note that, for all the compared methods, we always use the best possible hyperparameters to make sure that the comparison is fair. The baseline has exactly the same architecture and training settings as the one that MHE uses, and the only difference is an additional MHE regularization. For full-space MHE in hidden layers, we set $\lambda_h$ as 10 for all experiments. For half-space MHE in hidden layers, we set $\lambda_h$ as 1 for all experiments. For MHE in output layers, we set $\lambda_o$ as 1 for all experiments. We use $1e-5$ for the orthonormal regularization. If not otherwise specified, standard $\ell_2$ weight decay ($1e-4$) is applied to all the neural network including baselines and the networks that use MHE regularization. A very minor issue for the hyperparameters $\lambda_h$ is that it may increase as the number of layers increases, so we can potentially further divide the hyperspherical energy for the hidden layers by the number of layers. It will probably change the current optimal hyperparameter setting by a constant multiplier. For notation simplicity, we do not explicitly write out the weight decay term in the loss function in the main paper. Note that, all the neuron weights in the neural networks used in the paper are not normalized (unless otherwise specified), but the MHE will normalize the neuron weights while computing the regularization loss. As a result, *MHE does not need to modify any component of the original neural networks, and it can simply be viewed as an extra regularization loss that can boost the performance*. Because half-space variants can only applied to the hidden layers, both original MHE and its half-space version apply the full-space MHE to the output layer by default. The difference between MHE and half-space MHE are only in the regularization for the hidden layers.

**Class-imbalance learning.** There are 50000 training images in the original CIFAR-10 dataset, with 5000 images per class. For the single class imbalance setting, we keep original images of class

1-9 and randomly throw away 90% images of class 0. The total number of training images in this setting is 45500. For the multiple class imbalance setting, we set the number of each class equals to $500 \times (\text{class\_index} + 1)$. For instance, class 0 has 500 images, class 1 has 1000 images and class 9 has 5000 images. The total number of training images in this setting is 27500. Note that, both half-space MHE and half-space A-MHE in Table 7 and Table 8 mean that the half-space variants have been applied to the hidden layers. For the output layer (*i.e.*, classifier neurons), only full-space MHE can be used.

**SphereFace+.** SphereFace+ uses the same face detection and alignment method [61] as SphereFace [28]. The testing protocol on LFW and MegaFace is also the same as SphereFace. We use exactly the same preprocessing as in the SphereFace repository. Detailed network architecture settings of SphereFace-20 and SphereFace-64 can be found in [28]. Specifically, we use full-space MHE with Euclidean distance and $s = 2$ in the output layer. Essentially, we treat MHE as an additional loss function which aims to enlarge the inter-class angular distance of features and serves a complementary role to the angular softmax in SphereFace. Note that, for the results of CosineFace [51], we directly use the results (with the same training settings and without using feature normalization) reported in the paper. Since ours also does not perform feature normalization, it is a fair comparison. With feature normalization, we find that the performance of SphereFace+ will also be improved significantly. However, feature normalization makes the results more tricky, because it will involve another hyperparameter that controls the projection radius of feature normalization.

In order to reduce the training difficulty, we adopt a new training strategy. Specifically, we first train a model using the original SphereFace, and then use the new loss function proposed in Eq. 8 to finetune the pretrained SphereFace model. Note that, only the results for face recognition are obtained using this training strategy.

# B Proof of Theorem 1 and Theorem 2

Theorem 1 and Theorem 2 are natural results from classic potential theory [25] and spherical configuration [11, 23, 8]. We discuss the asymptotic behavior ($N \to \infty$) in three cases: $0 < s < d$, $s = d$, and $s > d$. We first write the energy integral as

$$I_s(\mu) = \iint_{\mathbb{S}^d \times \mathbb{S}^d} \|\boldsymbol{u} - \boldsymbol{v}\|^{-s} d\mu(\boldsymbol{u}) d\mu(\boldsymbol{v}), \tag{9}$$

which is taken over all probability measure $\mu$ supported on $\mathbb{S}^d$. With $0 < s < d$, $I_s(\mu)$ is minimal when $\mu$ is the spherical measure $\sigma^d = \mathcal{H}^d(\cdot)|_{\mathbb{S}^d}/\mathcal{H}^d(\mathbb{S}^d)$ on $\mathbb{S}^d$, where $\mathcal{H}^d(\cdot)$ denotes the $d$-dimensional Hausdorff measure. When $s \geq d$, $I_s(\mu)$ becomes infinity, which therefore requires different analysis.

First, the classic potential theory [25] can directly give the following results for the case where $0 < s < d$:

**Lemma 1.** *If $0 < s < d$,*

$$\lim_{N \to \infty} \frac{\varepsilon_{s,d}(N)}{N^2} = I_s\left(\frac{\mathcal{H}^d(\cdot)|_{\mathbb{S}^d}}{\mathcal{H}^d(\mathbb{S}^d)}\right), \tag{10}$$

*where $I_s$ is defined in the main paper. Moreover, any sequence of optimal hyperspherical $s$-energy configurations $(\hat{\boldsymbol{W}}_N^\star)|_2^\infty \subset \mathbb{S}^d$ is asymptotically uniformly distributed in the sense that for the weak-star topology measures,*

$$\frac{1}{N} \sum_{\boldsymbol{v} \in \hat{\boldsymbol{W}}_N^\star} \delta_{\boldsymbol{v}} \to \sigma^d, \quad \text{as } N \to \infty \tag{11}$$

*where $\delta_{\boldsymbol{v}}$ denotes the unit point mass at $\boldsymbol{v}$, and $\sigma^d$ is the spherical measure on $\mathbb{S}^d$.*

which directly concludes Theorem 1 and Theorem 2 in the case of $0 < s < d$.

For the case where $s = d$, we have from [23, 8] the following results:

**Lemma 2.** *Let $\mathcal{B}^d := \bar{B}(0,1)$ be the closed unit ball in $\mathbb{R}^d$. For $s = d$,*

$$\lim_{N \to \infty} \frac{\varepsilon_{s,d}(N)}{N^2 \log N} = \frac{\mathcal{H}^d(\mathcal{B}^d)}{\mathcal{H}^d(\mathbb{S}^d)} = \frac{1}{d} \frac{\Gamma(\frac{d+1}{2})}{\sqrt{\pi}\Gamma(\frac{d}{2})}, \tag{12}$$

*and any sequence $(\hat{\boldsymbol{W}}_N^\star)|_2^\infty \subset \mathbb{S}^d$ of optimal $s$-energy configurations satisfies Eq. 11.*

which concludes the case of $s = d$. Therefore, we are left with the case where $s > d$. For this case, we can use the results from [11]:

**Lemma 3.** *Let $A \subset \mathbb{R}^d$ be compact with $\mathcal{H}_d(A) > 0$, and $\tilde{W}_N = \{x_{k,N}\}_{i=1}^N$ be a sequence of asymptotically optimal $N$-point configurations in $A$ in the sense that for some $s > d$,*

$$\lim_{N \to \infty} \frac{\boldsymbol{E}_s(\tilde{W}_N)}{N^{1+s/d}} = \frac{C_{s,d}}{\mathcal{H}^d(A)^{s/d}} \tag{13}$$

*or*

$$\lim_{N \to \infty} \frac{\boldsymbol{E}_s(\tilde{W}_N)}{N^2 \log N} = \frac{\mathcal{H}^d(\mathcal{B}^d)}{\mathcal{H}^d(A)}. \tag{14}$$

*where $C_{s,d}$ is a finite positive constant independent of $A$. Let $\delta_x$ be the unit point mass at the point $x$. Then in the weak-star topology of measures we have*

$$\frac{1}{N} \sum_{i=1}^N \delta_{x_{i,N}} \to \frac{\mathcal{H}^d(\cdot)|_A}{\mathcal{H}^d(A)}, \quad as N \to \infty. \tag{15}$$

The results naturally prove the case of $s > d$. Combining these three lemmas, we have proved Theorem 1 and Theorem 2.

# C Understanding MHE from Decoupled View

Inspired by decoupled networks [27], we can view the original convolution as the multiplication of the angular function $g(\theta) = \cos(\theta)$ and the magnitude function $h(\|\boldsymbol{w}\|, \|\boldsymbol{x}\|) = \|\boldsymbol{w}\| \cdot \|\boldsymbol{x}\|$:

$$
\begin{aligned}
f(\boldsymbol{w}, \boldsymbol{x}) &= h(\|\boldsymbol{w}\|, \|\boldsymbol{x}\|) \cdot g(\theta) \\
&= \big(\|\boldsymbol{w}\| \cdot \|\boldsymbol{x}\|\big) \cdot \big(\cos(\theta)\big)
\end{aligned}
\tag{16}
$$

where $\theta$ is the angle between the kernel $\boldsymbol{w}$ and the input $\boldsymbol{x}$. From the equation above, we can see that the norm of the kernel and the direction (*i.e.*, angle) of the kernel affect the inner product similarity differently. Typically, weight decay is to regularize the kernel by minimizing its $\ell_2$ norm, while there is no regularization on the direction of the kernel. Therefore, MHE is able to complete this missing piece by promoting angular diversity. By combining MHE to a standard neural networks (*e.g.*, CNNs), the regularization term becomes

$$
\mathcal{L}_{\text{reg}} = \underbrace{\lambda_{\text{w}} \cdot \frac{1}{\sum_{j=1}^{L} N_j} \sum_{j=1}^{L} \sum_{i=1}^{N_j} \|\boldsymbol{w}_i\|}_{\text{Weight decay: regularizing the magnitude of kernels}} + \underbrace{\lambda_{\text{h}} \cdot \sum_{j=1}^{L-1} \frac{1}{N_j(N_j-1)} \{\boldsymbol{E}_s\}_j + \lambda_{\text{o}} \cdot \frac{1}{N_L(N_L-1)} \boldsymbol{E}_s(\hat{\boldsymbol{w}}_i^{\text{out}}|_{i=1}^{c})}_{\text{MHE: regularizing the direction of kernels}}
\tag{17}
$$

where $\boldsymbol{x}_i$ is the feature of the $i$-th training sample entering the output layer, $\boldsymbol{w}_i^{\text{out}}$ is the classifier neuron for the $i$-th class in the output fully-connected layer and $\hat{\boldsymbol{w}}_i^{\text{out}}$ denotes its normalized version. $\{\boldsymbol{E}_s\}_i$ denotes the hyperspherical energy for the neurons in the $i$-th layer. $c$ is the number of classes, $m$ is the batch size, $L$ is the number of layers of the neural network, and $N_i$ is the number of neurons in the $i$-th layer. $\boldsymbol{E}_s(\hat{\boldsymbol{w}}_i^{\text{out}}|_{i=1}^{c})$ denotes the hyperspherical energy of neurons $\{\hat{\boldsymbol{w}}_1^{\text{out}}, \cdots, \hat{\boldsymbol{w}}_c^{\text{out}}\}$ in the output layer. $\lambda_{\text{w}}$, $\lambda_{\text{h}}$ and $\lambda_{\text{o}}$ are weighting hyperparameters for these three regularization terms.

From the decoupled view, we can see that MHE is actually very meaningful in regularizing the neural networks, and it also serves as a complementary role to weight decay. According to [27] (using classifier neurons as an intuitive example), weight decay is used to regularize the intra-class variation, while MHE is used to regularize the inter-class semantic difference. In such sense, MHE completes an important missing piece for the standard neural networks by regularizing the directions of neurons (*i.e.*, kernels). In contrast, the standard neural networks only have weight decay as a regularization for the norm of neurons.

Weight decay can help to prevent the network from overfitting and improve the generalization. Similarly, MHE can serve as a similar role, and we argue that MHE is very likely to be more crucial than weight decay in avoiding overfitting and improving generalization. Our intuition comes from SphereNets [30] which shows that the magnitude of kernels is not important for object recognition. Therefore, the directions of the kernels are directly related to the semantic discrimination of the neural networks, and MHE is designed to regularize the directions of kernels by imposing the hyperspherical diversity. To conclude, MHE provides a novel hyperspherical perspective for regularizing neural networks.

# D  Weighted MHE

In this section, we do a preliminary study for weighted MHE. To be clear, weighted MHE is to compute MHE with neurons with different fixed weights. Taking Euclidean distance MHE as an example, weighted MHE can be formulated as:

$$\boldsymbol{E}_{s,d}(\beta_i\hat{\boldsymbol{w}}_i|_{i=1}^N) = \sum_{i=1}^N \sum_{j=1,j\neq i}^N f_s\big(\|\beta_i\hat{\boldsymbol{w}}_i - \beta_j\hat{\boldsymbol{w}}_j\|\big) = \begin{cases} \sum_{i\neq j}\|\beta_i\hat{\boldsymbol{w}}_i - \beta_j\hat{\boldsymbol{w}}_j\|^{-s}, & s > 0 \\ \sum_{i\neq j}\log\big(\|\beta_i\hat{\boldsymbol{w}}_i - \beta_j\hat{\boldsymbol{w}}_j\|^{-1}\big), & s = 0 \end{cases},$$

(18)

where $\beta_i$ is a constant weight for the neuron $\boldsymbol{w}_i$. We perform a toy experiment to see how these weights $\beta_i$ can affect the neuron distribution on 3-dimensional sphere. Specifically, we follow the same setting as Fig. 1, and apply weighted MHE to 10 normalized vectors in 3-dimensional space. We experiment two settings: (1) only one neuron $\boldsymbol{w}_1$ has different weight $\beta_1$ than the other 9 neurons; (2) two neurons $\boldsymbol{w}_1, \boldsymbol{w}_2$ have different weight $\beta_1, \beta_2$ than the other 8 neurons. For the first setting, we visualize the cases where $\beta_1 = 1, 2, 4, 10$ and $\beta_i = 1, 10 \geq i \geq 2$. The visualization results are shown in Fig. 5. For the second setting, we visualize the cases where $\beta_1 = \beta_2 = 1, 2, 4, 10$ and $\beta_i = 1, 10 \geq i \geq 3$. The visualization results are shown in Fig. 6. In these visualization experiments, we only use the gradient of weighted MHE to update the randomly initialized neurons. Note that, for all experiments, the random seed is fixed.

(a) β₁=1        (b) β₁=2        (c) β₁=4        (d) β₁=10

Figure 5: The visualization of normalized neurons after applying weighted MHE in the first setting. The blue-green square dots denote the trajectory (history of the iterates) of neuron $\boldsymbol{w}_1$ with $\beta_1 = 1, 2, 4, 10$, while the red dots denote the neurons with $\beta_i = 1, i \neq 1$. The final neuron $\boldsymbol{w}_1$ is connected to the origin with a solid blue line. The dash line is used to connected the trajectory.

(a) β₁=β₂=1        (b) β₁=β₂=2        (c) β₁=β₂=4        (d) β₁=β₂=10

Figure 6: The visualization of normalized neurons after applying weighted MHE in the second setting. The blue-green square dots denote the trajectory of neuron $\boldsymbol{w}_1$ with $\beta_1 = 1, 2, 4, 10$, the pure green square dots denote the trajectory of neuron $\boldsymbol{w}_2$ with $\beta_2 = 1, 2, 4, 10$, and the red dots denote the neurons with $\beta_i = 1, i \neq 1, 2$. The final neurons $\boldsymbol{w}_1$ and $\boldsymbol{w}_2$ are connected to the origin with a solid blue line and a solid green line, respectively. The dash line is used to connected the trajectory.

From both Fig. 5 and Fig. 6, one can observe that the neurons with larger $\beta$ tend to be more "fixed" (unlikely to move), and the neurons with smaller $\beta$ tend to move more flexibly. This can also be interpreted as the neurons with larger $\beta$ being more important. Such phenomena indicate that we can control the flexibility of the neurons under the learning dynamics of MHE. There is one scenario where weighted MHE may be very useful. Suppose we have known that some neurons are already well learned (*e.g.*, some filters from a pretrained model) and we do not want these neurons to be updated dramatically, then we can use the weighted MHE and set a larger $\beta$ for these neurons.

# E  Regularizing SphereNets with MHE

SphereNets [30] are a family of network networks that learns on hyperspheres. The filters in SphereNets only focus on the hyperspherical (*i.e.*, angular) difference. One can see that the intuition of SphereNets well matches that of MHE, so MHE can serve as a natural and effective regularization for SphereNets. Because SphereNets throw away all the magnitude information of filters, the weight decay can no longer serve as a form of regularization for SphereNets, which makes MHE a very useful regularization for SphereNets. Originally, we use the orthonormal regularization $\|\boldsymbol{W}^\top \boldsymbol{W} - \boldsymbol{I}\|_F^2$ to regularize SphereNets, where $\boldsymbol{W}$ is the weight matrix of a layer with each column being a vectorized filter and $\boldsymbol{I}$ is an identity matrix. We compare MHE, half-space MHE and orthonormal regularization for SphereNets. In this section, all the SphereNets use the same architecture as the CNN-9 in Table 11, the training setting is also the same as CNN-9. We only evaluate SphereNets with cosine SphereConv. Note that, $s = 0$ is actually the logarithmic hyperspherical energy (a relaxation of the original hyperspherical energy). From Table 13, we observe that SphereNets with MHE can outperform both the SphereNet baseline and SphereNets with the orthonormal regularization, showing that MHE is not only effective in standard CNNs but also very suitable for SphereNets.

| Method | CIFAR-10 | | | CIFAR-100 | | |
|---|---|---|---|---|---|---|
| | $s=2$ | $s=1$ | $s=0$ | $s=2$ | $s=1$ | $s=0$ |
| MHE | **5.71** | 5.99 | **5.95** | 27.28 | **26.99** | 27.03 |
| Half-space MHE | 6.12 | 6.33 | 6.31 | 27.17 | 27.77 | 27.46 |
| A-MHE | 5.91 | 5.98 | 6.06 | **27.07** | 27.27 | **26.70** |
| Half-space A-MHE | 6.14 | **5.87** | 6.11 | 27.35 | 27.68 | 27.58 |
| SphereNet with Orthonormal Reg. | 6.13 | | | 27.95 | | |
| SphereNet Baseline | 6.37 | | | 28.10 | | |

Table 13: Testing error (%) of SphereNet with different MHE on CIFAR-10/100.

# F   Improving AM-Softmax with MHE

We also perform some preliminary experiments for applying MHE to additive margin softmax loss [49] which is a recently proposed well-performing objective function for face recognition. The loss function of AM-Softmax is given as follows:

$$\mathcal{L}_{\text{AMS}} = -\frac{1}{n} \sum_{i=1}^{n} \log \frac{e^{s \cdot \left( cos\theta_{(\boldsymbol{x}_i, \boldsymbol{w}_{y_i})} - m_{\text{AMS}} \right)}}{e^{s \cdot \left( cos\theta_{(\boldsymbol{x}_i, \boldsymbol{w}_{y_i})} - m_{\text{AMS}} \right)} + \sum_{j=1, j \neq y_i}^{c} e^{s \cdot cos\theta_{(\boldsymbol{x}_i, \boldsymbol{w}_j)}}} \tag{19}$$

where $y_i$ is the label of the training sample $x_i$, $n$ is the mini-batch size, $m_{\text{AMS}}$ is the hyperparameter that controls the degree of angular margin, and $\theta_{(\boldsymbol{x}_i, \boldsymbol{w}_j)}$ denotes the angle between the training sample $\boldsymbol{x}_i$ and the classifier neuron $\boldsymbol{w}_j$. $s$ is the hyperparameter that controls the projection radius of feature normalization [50, 49]. Similar to our SphereFace+, we combine full-space MHE to the output layer to improve the inter-class feature separability. It is essentially following the same intuition of SphereFace+ by adding an additional loss function to AM-Softmax loss.

**Experiments.** We perform a preliminary experiment to study the benefits of MHE for improving AM-Softmax loss. We use the SphereFace-20 network and trained on CASIA-WebFace dataset (training settings are exactly the same as SphereFace+ in the main paper and [28]). The hyperparameters $s, m_{\text{AMS}}$ for AM-Softmax loss exactly follow the best setting in [49]. AM-Softmax achieves 99.26% accuracy on LFW, while combining MHE with AM-Softmax yields 99.37% accuracy on LFW. Such performance gain is actually very significant in face verification, which further validates the superiority of MHE.

# G  Improving GANs with MHE

We propose to improve the discriminator of GANs using MHE. It has been pointed out in [34] that the function space from which the discriminators are learned largely affects the performance of GANs. Therefore, it is of great importance to learn a good discriminator for GANs. As a recently proposed regularization to stabilize the training of GANs, spectral normalization (SN) [34] encourages the Lipschitz constant of each layer's weight matrix to be one. Since MHE exhibits significant performance gain for CNNs as a regularization, we expect MHE can also improve the training of GANs by regularizing its discriminator. As a result, we perform a preliminary evaluation on applying MHE to GANs.

Specifically, for all methods except WGAN-GP [9], we use the standard objective function for the adversarial loss:

$$V(G, D) := \mathbb{E}_{\boldsymbol{x} \sim q_{\text{data}}(\boldsymbol{x})}[\log D(\boldsymbol{x})] + \mathbb{E}_{\boldsymbol{z} \sim p(\boldsymbol{z})}[\log(1 - D(G(\boldsymbol{z})))], \tag{20}$$

where $\boldsymbol{z} \in \mathbb{R}^{d_z}$ is a latent variable, $p(\boldsymbol{z})$ is the normal distribution $\mathcal{N}(0, I)$, and $G : \mathbb{R}^{d_z} \to \mathbb{R}^{d_0}$ is a deterministic generator function. We set $d_z$ to 128 in all the experiments. For the updates of $G$, we used the alternate cost proposed by [7] $-\mathbb{E}_{\boldsymbol{z} \sim p(\boldsymbol{z})}[\log(D(G(\boldsymbol{z})))]$ as used in [7, 52]. For the updates of $D$, we used the original cost function defined in Eq. (20).

Recall from [34] that spectral normalization normalizes the spectral norm of the weight matrix $\boldsymbol{W}$ such that it makes the Lipschitz constraint $\sigma(\boldsymbol{W})$ to be one:

$$\bar{\boldsymbol{W}}_{\text{SN}}(\boldsymbol{W}) := \frac{\boldsymbol{W}}{\sigma(\boldsymbol{W})}. \tag{21}$$

We apply MHE to the discriminator of standard GANs (with the original loss function in [7]) for image generation on CIFAR-10. In general, our experimental settings and training strategies (including architectures in Table 15) exactly follow spectral normalization [34]. For MHE, we use the half-space variant with Euclidean distance (Eq. (1)). We first experiment regularizing the discriminator using MHE alone, and it yields comparable performance to SN and orthonormal regularization. Moreover, we also regularize the discriminator simultaneously using both MHE and SN, and it can give much better results than using either SN or MHE alone. The results in Table 14 show that MHE is potentially very useful for training GANs.

| Method | Inception score |
|---|---|
| Real data | 11.24±.12 |
| Weight clipping | 6.41±.11 |
| GAN-gradient penalty (GP) | 6.93±.08 |
| WGAN-GP [9] | 6.68±.06 |
| Batch Normalization [20] | 6.27±.10 |
| Layer Normalization [2] | 7.19±.12 |
| Weight Normalization [39] | 6.84±.07 |
| Orthonormal [4] | 7.40±.12 |
| SN-GANs [34] | **7.42**±.08 |
| MHE (ours) | 7.32±.10 |
| MHE + SN [34] (ours) | **7.59**±.08 |

Table 14: Inception scores with unsupervised image generation on CIFAR-10.

## G.1 Network Architecture for GAN

We give the detailed network architectures in Table 15 that are used in our experiments for the generator and the discriminator.

Table 15: Our CNN architectures for image Generation on CIFAR-10. The slopes of all leaky ReLU (lReLU) functions in the networks are set to 0.1.

| $z \in \mathbb{R}^{128} \sim \mathcal{N}(0, I)$ |
|---|
| dense $\rightarrow M_g \times M_g \times 512$ |
| 4×4, stride=2 deconv. BN 256 ReLU |
| 4×4, stride=2 deconv. BN 128 ReLU |
| 4×4, stride=2 deconv. BN 64 ReLU |
| 3×3, stride=1 conv. 3 Tanh |

(a) Generator ($M_g = 4$ for CIFAR10).

| RGB image $x \in \mathbb{R}^{M \times M \times 3}$ |
|---|
| 3×3, stride=1 conv 64 lReLU |
| 4×4, stride=2 conv 64 lReLU |
| 3×3, stride=1 conv 128 lReLU |
| 4×4, stride=2 conv 128 lReLU |
| 3×3, stride=1 conv 256 lReLU |
| 4×4, stride=2 conv 256 lReLU |
| 3×3, stride=1 conv. 512 lReLU |
| dense $\rightarrow 1$ |

(b) Discriminator ($M = 32$ CIFAR10).

## G.2 Comparison of Random Generated Images

We provide some randomly generated images for comparison between baseline GAN and GAN regularized by both MHE and SN. The generated images are shown in Fig. 7.

Dataset　　　　　　　　　　Baseline GAN　　　　　　　GAN with MHE and SN

Figure 7: Results of generated images.

# H   More Results on Class-imbalance Learning

## H.1   Class-imbalance learning on CIFAR-100

We perform additional experiments on CIFAR-100 to further validate the effectiveness of MHE in class-imbalance learning. In the CNN used in the experiment, we only apply MHE (*i.e.*, full-space MHE) to the output layer, and use MHE or half-space MHE in the hidden layers. In general, the experimental settings are the same as the main paper. We still use CNN-9 (which is a 9-layer CNN from Table 11) in the experiment. Slightly differently from CIFAR-10 in the main paper, the two data imbalance settings on CIFAR-100 include 1) 10-class imbalance (denoted as Single in Table 16) - All classes have the same number of images but 10 classes (index from 0 to 9) have significantly less number (only 10% training samples compared to the other normal classes), and 2) multiple class imbalance (denoted by Multiple in Table 16) - The number of images decreases as the class index decreases from 99 to 0. For the multiple class imbalance setting, we set the number of each class equals to $5 \times (\text{class\_index} + 1)$. Experiment details are similar to the CIFAR-10 experiment, which is specified in Appendix A. The results in Table 16 show that MHE consistently improves CNNs in class-imbalance learning on CIFAR-100. In most cases, half-space MHE performs better than full-space MHE.

| Method | Single | Multiple |
|---|---|---|
| Baseline | 31.43 | 38.39 |
| Orthonormal | 30.75 | 37.89 |
| MHE | **29.30** | 37.07 |
| Half-space MHE | 29.40 | **36.52** |
| A-MHE | 30.16 | 37.54 |
| Half-space A-MHE | 29.60 | 37.07 |

Table 16: Error rate (%) on imbalanced CIFAR-100.

## H.2   2D CNN Feature Visualization

(a) CNN without MHE (Training Set)

(b) CNN with MHE (Training Set)

(c) CNN without MHE (Testing Set)

(d) CNN features with MHE (Testing Set)

Figure 8: 2D CNN features with or without MHE on both training set and testing set. The features are computed by setting the output feature dimension as 2, similar to [29]. Each point denotes the 2D feature of a data point, and each color denotes a class. The red arrows are the classifier neurons of the output layer.

The experimental settings are the same as the main paper. We supplement the 2D feature visualization on testing set in Fig. 8. The visualized features on both training set and testing set well demonstrate the superiority of MHE in class-imbalance learning. In the CNN without MHE, the classifier neuron of the imbalanced training data is highly biased towards another class, and therefore can not be properly learned. In contrast, the CNN with MHE can learn uniformly distributed classifier neurons, which greatly improves the network's generalization ability.

# I More results of SphereFace+ on Megaface Challenge

We give more experimental results of SphereFace+ on Megaface challenge. The results in Table 17 evaluate SphereFace+ under different $m_{\text{SF}}$ and show that SphereFace+ consistently outperforms the SphereFace baseline. It indicates that MHE also enhances the verification rate on Megaface challenge. Our results of Identification Rate vs. Distractors Size and ROC curve are showed in Fig. 9 and Fig. 10, respectively.

| $m_{\text{SF}}$ | SphereFace | SphereFace+ |
|---|---|---|
| 1 | 42.46 | **52.02** |
| 2 | 71.79 | **80.94** |
| 3 | 76.34 | **80.58** |
| 4 | 82.56 | **83.39** |

Table 17: Megaface Verification Rate (%) of SphereFace+ under Res-20

Figure 9: Rank-1/Rank-10 Identification Performance on Megaface.

Figure 10: ROC Curve with 1M/10k Distractors on Megaface.