[Reviews · NeurIPS 2018]

Reviewer 1



[Summary] The over-parameterization nature of neural networks has an undesirable consequence that many units become highly correlated after model training. To reduce the redundancy in representation, this paper extends some recent work on diversity regularization in neural networks by proposing the so-called hyperspherical potential energy defined on the Euclidean distance or the angular distance, which, when minimized, helps to increase the diversity of the input weights of different neurons and hence reduce the redundancy. The regularized loss function contains two regularization terms corresponding to the hidden units and the output units, respectively. [Quality] Note that only the normalized weight vectors lie on the surface of the unit hypersphere. The unnormalized weight vectors generally do not. However, what actually contributes to the computation by each unit (neuron) in the neural network is the unnormalized weight vector. So two normalized weight vectors that are close to each other on the surface of the hypersphere may not actually be close when we consider the unnormalized weight vectors. How does this discrepancy affect the technical soundness of the proposed method? [Clarity] It is a bit misleading to say “the diversity of neurons” or “the neurons are more diverse”. I think by “neuron” you actually mean “the input weights of the neuron”. Figure 2 shows an example with only two neurons. Suppose there are many more neurons. How can we ensure that the virtual neurons all reside in the same half-space like that in Figure 1? It is said in Section 3.7 that previous methods based on orthogonality/angle-promoting regularization model diversity regularization at a more local level while the proposed method based on MHE regularization achieves it in a more top-down manner. I don’t understand what it means by “local level” and “top-down manner” here. Please elaborate. [Originality] There exist some previous works taking the orthogonality/angle-promoting regularization approach. The novelty of this work is in the MHE regularization terms. [Significance] The empirical results generally look promising. The theoretical analysis in Section 4 seems to imply that reducing the hyperspehical energy and hence increasing the diversity is good as it is closely related to improvement in generalization (Section 4.2). However, from the objective function in equation (4), it should be noted that the interplay between data fitting and hyperspherical energy minimization is crucial to achieving good performance. Among other things, finding the proper regularization parameters \lambda_h and \lambda_o is essential. Figure 3 shows the sensitivity analysis for CIFAR-100 with the CNN-9 network model by varying the value of the hyperparameters (I suppose it means both hyperparameters have the same value). What is not known is whether the hyperparameters are sensitive to the dataset and model used. Does changing either the dataset or the model require substantial effort in finding the right values for the hyperparameters? There are some language errors, e.g., L9: “as even as possible” L26: “that principled guides” L28: “Another stream of works seek to” L37: “geodesic distances … plays” L80: “class-imbalance” L116: “equivalent to optimize” L145: missing “.” after “redundancy” L218: “uniformly distribution” L267: “Results is given” L268: “both … outperforms” L278: “Both … performs” L298: “the our” L302: “can yields” L359: “complementary”, not “complimentary” L364: “as describe”; “MHE complete” Please proofread the paper more thoroughly. [Post-rebuttal comments] I have read the author feedback and the other two reviews. Regarding my comment on using unnormalized weight vectors, I do know the recent research on decoupling intra-class variation from inter-class variation [22-25]. My concern is that MHE (not angular-MHE) does not just focus on the angle when formulating the objective function. The formulation for angular-MHE is more correct. However, in the experiments, A-MHE is not consistently better than MHE. Deeper analysis and understanding of this is necessary. For my other questions, the author has generally provided satisfactory answers. I have increased my overall score from “5” (marginally below the acceptance threshold) to “6” (marginally above the acceptance threshold), but I don't think I am ready to vote for anything higher than “6”.

Reviewer 2



This paper proposes a minimum hyperspherical energy objective as a regularizer for removing the redundancy among the neurons for image classification. The idea is motivated by the Thomson problem in physics. Extensive experiments validate its effectiveness for different types of neural networks. However, I still have some concerns: i) The idea of drop out actually makes the idea robust to occlusion, and and it may help learn similar neurons for classification. In your experiments, if we block the image with some black blocks of different sizes, say10%\times, 30%\times, 50%\times, even70%\times of image width, will the performance of the proposed method still be good? In other words, is the proposed solution robust to the occlusion? ii) I would like to see the performance of the proposed method to other network architecture, like densenet, and GoogleNet.. iii) Please evaluate the proposed method on ImageNet. ImageNet is a more real and challenging dataset, it contains lots of occlusion. If the proposed method cannot be shown its effectiveness on ImageNet, I would like to doubt the effectivness for real data. Further, the improvement of the proposed solution is marginal on many datasets reported in the paper.

Reviewer 3



This paper focuses on the network regularization, and proposes a minimum hyperspherical energy objective with a few variants, which aims to vary the orientations of the neuron weights by minimizing the energy function. I think the idea is reasonable, but I have some concerns of this paper as follows: 1. The novelty of this paper should be clarified. As indicated in Eq.(3), the proposed objective seems to minimize the distance between every two neurons, which is similar with the existing works like [31] to minimize the cosine distance between any pair of feature vectors. Authors should give more explanation on the global, top-down manner of the proposed approach, comparing with the existing works. 2. Authors should consider more comprehensive comparison with existing works, where some state-of-the-art network regularization methods, including orthogonality/angle-promoting regularization should be compared to verify its effectiveness.